# Quantifying the Selective, Stochastic, and Complementary Drivers of Institutional Evolution in Online Communities

**DOI:** 10.3390/e24091185

**Published:** 2022-08-25

**Authors:** Qiankun Zhong, Seth Frey, Martin Hilbert

**Affiliations:** Department of Communication, University of California, Davis, CA 95616, USA

**Keywords:** price equation, bet-hedging, cultural evolution, online community

## Abstract

Institutions and cultures usually evolve in response to environmental incentives. However, sometimes institutional change occurs due to stochastic drivers beyond current fitness, including drift, path dependency, blind imitation, and complementary cooperation in fluctuating environments. Disentangling the selective and stochastic components of social system change enables us to identify the key features of long-term organizational development. Evolutionary approaches provide organizational science with abundant theories to demonstrate organizational evolution by tracking beneficial or harmful features. In this study, focusing on 20,000 Minecraft communities, we measure these drivers empirically using two of the most widely applied evolutionary models: the Price equation and the bet-hedging model. As a result, we find strong selection pressure on administrative and information rules, suggesting that their positive correlation with community fitness is the main reason for their frequency change. We also find that stochastic drivers decrease the average frequency of administrative rules. The result makes sense when viewed in the context of evolutionary bet-hedging. We show through the bet-hedging result that institutional diversity contributes to the growth and stability of rules related to information, communication, and economic behaviors.

## 1. Introduction

What are the main factors that drive institutional change? This is a central question across organizational theories. Indeed, the major forces behind institutional change can often be taxonomized: internal stability, external pressure, information transmission, institutional isomorphism, path dependency, and so on. Some of these factors are directly related to the payoff of implementation, whereas others are driven by stochastic forces. While stylized facts and intuition abound in this area, we have little empirical evidence due to a lack of both adequate data and general frameworks for comparing these sources of change and showing how they work together.

The migration of organizations to digital platforms has allowed researchers to obtain more adequate data, thanks to the digital footprint that online organizations inevitably leave behind. Traditionally, it was very difficult to obtain statistically significant samples from comparable organizations. Lab experiments with a large N of communities are expensive and often impractical. Natural experiments cannot ensure similar enough samples to accurately infer the effects of the variable in question. Digital trace data from online communities make it possible to monitor the intergenerational frequency changes in rules [1] because they provide fine-grained information about when rules were implemented, changed, and removed by and for thousands of similar online communities

The evolutionary framework adopted by researchers in various social science disciplines, including communication [2,3,4], economics [5,6,7,8], and sociology [9,10], provides theoretical and methodological support for the quantification of the resulting dynamics. In the past few decades, social science researchers have used concepts from biological evolution as an analogy to characterize the four main stages of institutional development: variation, selection, retention, and struggle [11]. This framework categorizes various institutional changes by the mechanism that drives them and provides explanations from both organizational and environmental perspectives. At the same time, the evolutionary framework provides tools to represent this analogy with mathematical relationships and explain the macro-dynamics based on a few first principles [12]. With empirical data, evolutionary models make it possible to quantify the strength of different drivers of institutional change and predict future developments.

One of the most comprehensive and successful models to describe the biological evolutionary process is the Price equation [13]. This equation partitions total evolutionary changes into two components: deterministic changes driven by natural selection and stochastic changes driven by all other forces, including adaptation, mismatch, drift, and biased transmission. The Price equation thus provides mathematical tools to separate selective and stochastic forces and reconcile different sources of change in a community [14,15].

In this study, we take the advantage of online community data to monitor rule changes among 20,000 Minecraft communities over two years. Online platforms including Wikipedia, Reddit, and Minecraft provide a great opportunity to study intergenerational changes of the modular institutional traits among thousands of small-scale communities. 

Using the Price equation, we are able to quantify how much of the observed community fitness is driven by natural selection and how much by stochastic forces that are not directly related to the success of the communities in question. We can also further explore the institutional structures of online communities in order to determine whether evolutionary forces are different for different types of rules. For example, do rules facilitate centralized, top-down communication driven by selective forces? Are stochastic forces more prevalent in rules regulating user behavior compared to rules regulating administrative behavior? Furthermore, is this result robust to changes in the environment? The bet-hedging method makes it possible to use information from an environment to match the frequency of rules, which produces a benchmark for a theoretically optimal rule distribution strategy. This method thus allows us to ask two questions. First, is the frequency change in one type of rule caused by this rule type only, or is it also influenced by other types of rules? Second, is the optimal distribution of rules consistent with the Price equation result? If the two models produce consistent results, we can conclude that the selection and stochasticity calculated through the Price equation are robust to changes and uncertainty in the environment; otherwise, we expect that environmental changes play a bigger role in the evolutionary dynamics of institutional changes.

As a result, we found that there are strong selective forces in the Minecraft environment that drives the frequency change of some rules, while at the same time, drift also exists among administrative rules, reducing their frequency. Additionally, the bet-hedging result suggests that communities need to subsidize other types of rules to achieve resilience against environmental fluctuation.

### 1.1. Institutional Change

The development of institutions has been a key research aspect of organizational studies [16,17,18]. To understand why and how institutions change, social science disciplines, including communication [3], sociology [10], and economics [7], have adopted an evolutionary framework to understand the dynamics of institutional development. Institutions as a set of rules to constrain behavior can be transmitted via communication processes and social learning [8,17]. An evolutionary approach to institutions allows us to examine both the processes involved in the origin, maintenance, and spread of specific rules as well as the complex ways in which different rules can interact to produce emergent properties at the populational level.

#### 1.1.1. The Arguments for Adaptive Selection 

Evolutionary frameworks in organizational studies focus on the natural selection of rules. The selection of rules is a process by which rule frequency increases or decreases as a result of the direct payoff resulting from the implementation of a particular rule. All selective processes are characterized by variation, heritability, and competition [11,19]. In an institutional context, rule variations arise across groups. With the variation and differential payoffs of rules, there should also be some form of competition between institutions regarding how the rules are beneficial for achieving organizational goals such as economic growth [20], political stability [21,22], successful localized management of common-pool resources [23], and long-term resilience [17,24,25]. Selective forces affecting institutions can occur under three conditions. First, groups with high payoff institutions outcompete other groups, replacing such groups or imposing their institutions on them [26]. For example, the rise of information and communication technologies (ICTs) has enabled a shift from group- to network-based societies because the latter derived greater benefits from ICTs [27]. Second, group members have high degrees of leverage to migrate to communities with better institutions at a low cost [28]. Banzhaf and Walsh provided empirical evidence that supports the notion that households “vote with their feet” for better institutions [29]. Third, certain institutions are more likely to be transmitted from one group to another. Zhong and Frey found that centralized rules are more likely to be transmitted than decentralized rules between online communities with overlapping members [30].

#### 1.1.2. Arguments for Stochasticity 

Although social science research that adopts an evolutionary framework mostly focuses on natural selection, many institutional changes are not driven by selective forces. Non-fitness-related changes are categorized as drift or stochastic forces [7]. Two major mechanisms in organizational research characterize this type of change in institutional settings: path dependency and institutional isomorphism. Path dependency refers to the process by which institutional development depends on a unique series of past events. The path cannot be retracted, nor can it be easily deflected. Path dependency can be explained through diverse mechanisms, including self-reinforcement [31,32], positive externalities [33], and lock-in [34]. Although institutional changes driven by path dependency can be beneficial for organizational success, for the time being, increased frequency is not related to organizational success. Institutional isomorphism [35] refers to the process whereby organizations borrow the routines, rules, and behavior from other organizations, regardless of possible mismatches between the adopted institution and the organizational context. Institutional isomorphism is explained through an organization’s internal bounded reality and the uncertainty or pressure of the external environment [36,37,38].

#### 1.1.3. Integrating and Disentangling Selective and Stochastic Forces

Even if natural selection is overemphasized when explaining organizational changes, the evolutionary framework has major benefits for studying institutional and organizational development. Among other things, it provides a formal theoretical framework based on first principles about how inherited traits will change over time under certain conditions. It is precisely these tools that let us articulate the relationship of natural selection to the many other evolutionary processes at work in social system changes. In social sciences, evolutionary explanations are often conflated with selective processes. Stochastic processes, although well studied in organizational studies, are usually not considered from the perspective of institutional organizations. The separation of the two main forces leads to some problems in identifying the true mechanisms of institutional changes. For example, institutional development driven by path dependency may also have direct benefits that are selected for due to competition with other institutions. Indeed, the rise of platforms, including Apple’s iOS and Google’s Android, enjoyed both builder and developer benefits at the beginning. However, lock-in benefits discourage the construction of gateways and thus force developers to commit to just one platform or to build and maintain multiple versions of the same product [39]. On the other hand, rules that can help achieve institutional goals may also be blindly borrowed by other groups without considering the context. Lowrey found that although there is blind isomorphism in the partnerships between newspapers and TV stations, the level of partnering is predicted by concrete benefits and the availability of resources [40]. These examples show that selective and stochastic forces are often conflated in institutional development. Focusing on one side of the story cannot provide a full picture of how different mechanisms work together in institutional changes. If we integrate different institutional change mechanisms, we will be able to answer an important question: how can we distinguish between the selective and stochastic forces in institutional development? In other words, how do we know whether the rule frequencies increase or decrease based on their contributions to the organizational goals or for other reasons, including path dependency and institutional isomorphism?

It has been difficult to answer this question empirically. First of all, institutions and other social-environmental processes, especially culture, are all endogenous. It is not straightforward to establish causal links between institutions and other factors [41,42,43]. Second, evolutionary processes can be analytically separated into discrete phases, but they are often linked in continuous feedback loops, making it difficult to map theoretical evolutionary stages to empirical data [11]. Variation provides sources for selection, but the selected traits after transmission and retention will, in turn, reduce variation among populations. As a result, evolutionary processes cannot help us evaluate any given moment in the process but rather form a dynamic system driven by different evolutionary forces. Third, selective and stochastic forces can vary over time. Institutions that were initially beneficial can end up reducing the rate of growth (e.g., due to lock-in effects). A clear identification strategy and longitudinal data are required to calculate their time-variant and average strength. Finally, it is difficult to quantify institutions and institutional changes due to their complex natures. There is no clear way to determine whether two institutions are comparable or whether we should take into account the interactions between rules within one institution.

In this paper, we address these difficulties using the Price equation and longitudinal data from online communities to disentangle different mechanisms in institutional evolution. We use a longitudinal online community dataset to make quantitative comparisons of thousands of organizations and apply the Price equation as a statistical strategy to make clear assessments of selective and stochastic forces.

### 1.2. The Price Equation

The Price equation [13] is one of the best-known biological evolutionary models. It is a theorem that represents any system of differential transmission [44]. In its original form in population biology, it provides a way to understand the effects that gene transmission and natural selection have on the frequency of alleles within each new generation of a population. Due to its abstract mathematical articulation, the Price equation is applied broadly in anthropology and economics [14]. Within the evolutionary framework we have described, it is reasonable to also apply the Price equation to organizational studies. This equation partitions total evolutionary change into two components: the abstract expression of natural selection (selective forces) and all other evolutionary processes (stochastic forces). The two pieces of the Price equation together can represent multiple evolutionary forces such as natural selection, shift, and biased cultural transmission. One general form of the Price Equation is
$$\Delta z=COV[\frac{w_{i}}{w},z_{i}]+E[\delta_{i}]$$
where $w_{i}$ refers to the direct fitness-related change in community *i*, associated with a cultural trait. In situations where we can draw direct causal links between a cultural trait and a change in fitness, $w_{i}$ can be interpreted as the payoff of the cultural trait; $z_{i}$ refers to the frequency of the trait in community *i*, and $\delta_{i}$ refers to the random change of the trait frequency in community *i*. This equation establishes that the fitness-correlated selective forces ($COV\left[\frac{w_{i}}{w},z_{i}\right]$) and the fitness-uncorrelated stochastic forces ($E\left[\delta_{i}\right]$) contribute to the frequency of change of a cultural trait, here designated as Δz.

With the theoretical mapping from biological to institutional evolution, we can use the Price equation to estimate the selective and stochastic forces influencing institutional changes at a rule level. 

### 1.3. Online Communities

Longitudinal data from online communities make it possible to quantify institutional changes and extract the measures in order to apply the Price equation empirically.

Online platforms including Wikipedia, the discussion platform Reddit, and the game Minecraft offer a meta-population of online communities. Such large-scale groups make it possible to compare the institutions of thousands of communities within the same macro environment and cultural context [45,46]. The communities within the same platform often face the same collective action problems and pursue the same organizational goals, allowing for meaningful comparisons of institutions.

In recent years, through methods including API and web scraping, we have been able to acquire longitudinal data on online communities and study the long-term institutional development of many thousands of groups at the same time [45,47,48,49,50,51]. In this research, we monitored over 20,000 Minecraft servers, which allow various user activities, including building with blocks, gathering resources, and interacting with each other. The servers thus function as communities with which users can engage. The Minecraft environment hosts millions of communities that compete for scarce physical and virtual resources and struggle for the same organizational goal—to recruit and retain members. The same collective problems and goals they face put them under selection pressure, whereas various choices administrators and community members have granted them space for stochastic drift. We collected data on the rules each community had implemented over two years. Modular rule sets, called “plugins” in the Minecraft world, provide a standardized means by which to quantify institutions and set a unit of analysis at the rule level. The plugin types can then be used to measure institutional traits. By calculating the frequency change and variance of one type of plugin, we were able to apply the Price equation in an institutional setting.

Using the Price equation and online community data, we try to determine:

**RQ1:** 
*What are the selective and stochastic forces that drive frequency changes in different kinds of rules in online communities?*


### 1.4. Time Variance and Institutional Diversity

Environmental fluctuations exogenous to culture and institutions have a large influence on cultural and institutional evolution. The frequency and intensity of environmental changes affect which type of cultural and institutional trait is selected and stabilized in the long run. For example, Roger’s model explains how conformity evolves only in situations where environmental changes are not frequent [52]. Giuliano and Nunn used a set of historical data to determine that populations that experience more cross-generational temperature instability attribute less importance to traditional values [53]. Richerson and Boyd attributed the emergence of cumulative culture to climate change in the late Pleistocene [12].

In the case of Minecraft, the software environment and version changes may cause changes in the payoff of implementing one type of rules and influence the evolutionary trajectory. The environment in Minecraft can therefore influence both selective and stochastic forces. For example, when the overall online community environment becomes more unpredictable or unstable, it is possible that institutions with decentralized rules that promote peer interactions will be more likely to be selected for comparisons with centralized rules that reinforce top-down hierarchies [54]. At the same time, an uncertain environment may increase blind imitation [35], leading to stochastic institutional changes. Thus:

**RQ2:** 
*Is the evolution trajectory of rules influenced by environmental changes in online communities?*


So far, we have considered how single institutional traits (rules) evolve in various environments. However, oftentimes, organizational development relies on complementary rules which function together. Ostrom proposed the Institutional Analysis and Development framework to analyze various social institutions and provided empirical evidence of the benefits of institutional diversity to robust, self-organized institutions [23]. Page provided evidence that supports the benefits of diversity in complex systems, especially in response to external shocks and internal adaptations [25]. In Minecraft, there are four types of meaningful rules related to governance. However, when we zoom in and only focus on a single type of rule, the Price equation forces us to include the influence of other types of rules based on stochastic forces and environmental factors. Whether other types of rules can interact with one particular type to operate together on institutional evolution through rule diversity requires further analysis. Motivated by Ostrom’s and Page’s theory, we ask:

**RQ3:** 
*Is the evolution of a single type of rule influenced by rule diversity among online communities?*


## 2. Materials and Methods

### 2.1. Data

We collected longitudinal plugin implementation data from 370,000 Minecraft servers through bihourly API queries between November 2014 and November 2016. After filtering out servers that were disconnected for the duration of data collection (~220,000), those that did not survive for at least a month (~70,000), and those that did not report full governance information (~75,000), we ended up with a sample of 14,859 servers (we address the limitation resulting from this data deletion process in the Limitations section).

In Minecraft, plugins are modular programs that administrators can install on their servers to automatically implement rules and other institutional constructs (See Appendix A for a detailed description of plugins). In the digital world, code is the law [55]. By mixing and matching plugins, Minecraft server administrators establish formal institutions to maintain community survival and achieve community success. The Minecraft community has developed almost 20,000 plugins listed under 16 categories, among which Frey and Sumner concluded that four rule types were directly related to governance: top-down administration, information broadcasting, communication, and economy [56]. Administration rules enhance administrator control over community and user behavior. Informational rules facilitate information sharing from administrators to users. Communication rules improve communication among players. Finally, economic rules protect private property and enable trade. To quantify institutional changes and analyze the evolution process in Minecraft, we used this classification to categorize the plugins. To fit the Price Equation, we summarize community-level data at the unit of one month. As the median “lifespan” of a server is 9 weeks, this aggregation provides an appropriate timescale to capture the dynamics of intra- and inter-generational cultural transmission.

We took a community as an organism and the share of rules (plugins) as the institutional trait or cultural variant it displays. In the large group of communities in Minecraft, we saw that different communities exhibited different institutional traits (cultural variants) which constituted the overall distribution of institutional traits in this setting. 

In Minecraft, we do not know if after a community dies, the governing knowledge is retained by its members to pass on to the next generation. In this sense, we could not strictly follow a genetic inheritance model. However, when a new community starts, to maintain community survival and deal with collective action problems, the community administrators need to learn, either socially from other communities or independently from the environment, in order to establish their institutional traits and governing style. The process of learning can be seen as cultural transmission that changes the overall distribution of rule shares.

Different forces contribute to distribution changes. Selective forces act on Minecraft communities in two ways. First, communities that employ a governance style that is beneficial for community survival will last longer. For example, if administrative rules are the most beneficial for community survival, communities that employ a large share of administrative rules will last longer. In contrast, communities that employ a governance style with a small share of administrative rules will die out faster. This differential survival rate of different governing styles will lead to a shift in the overall distribution of administrative rules. Second, communities that employ a governance style that is beneficial for community success are more likely to be copied by other communities. The spread of successful governance styles can also change the overall distribution of rule shares.

Stochastic forces also act on Minecraft communities in two major ways. First, communities blindly learn from other communities. When the learning is not led by success bias but rather by proximity or uncertainty, this type of copying will lead to drift in the overall distribution. Second, when players cultivate cultural preferences for specific governance styles, they are likely to spread these rule shares to other communities they migrate to [30].

Additionally, mutation provides additional variation for selection. In Minecraft, the introduction of new plugins or individual learning to establish new governance styles can be seen as a form of mutation.

Administrators do not know whether their implementation of one type of rule instead of another is due to selective or stochastic forces. Likewise, we cannot accurately identify selective or stochastic forces from the interactions among communities. What is available in this dataset is the collective pattern of rule distribution. The advantage of using the Price equation is that, in this case, it statistically isolates selective forces from stochastic forces without specifying a large number of possible mechanisms behind each basic process.

### 2.2. Price Equation 

The Price Equation provides a powerful generalization of the forces contributing to evolutionary changes in institutional structures—analogous in our framework to the biological traits of online communities—which correspond to individual organisms. Here, we demonstrate one possible decomposition of the Price Equation that focuses mainly on distinguishing among the strength of the correlation between the relative growth of a population (‘fitness’), the presence of a certain community trait (‘selection’ based on that trait), and the strength of other stochastic fluctuations.

Consider the Minecraft environment, which consists of multiple servers, each of which is indexed by *i* (N = 13,859 servers). Within each community, we identified four categories of rules: administrative, informational, communication, and economics (see Figure 1a). Taking administrative rules as an example, the relative frequency of administrative rules in community *i* is $z_{i}=r_{i}/R_{i}$, where $r_{i}$ is the number of administrative rules, and $R_{i}$ is the total number of rules in community *i*. Some communities might have many rules but a small population (see Figure 1b), so we considered it useful to create a measure of rule fraction weighted by membership population size. In most cultural evolution work, in order to estimate the fraction of a cultural trait within a population, researchers either calculate the number of individuals that carry a specific cultural trait weighted by the overall population size [57] or the number the artifacts of a kind of cultural feature weighted by the production size [58]. It is tricky to apply this kind of calculation to organizational and institutional evolution because conceptually, the individuals in the organization do not directly carry the cultural trait and the total number of rules does not reflect the “production size” of the community. Therefore, to establish the connection between the rule frequency to organizational size and to make sense of differential cultural transmission, we used the frequency of administrative rules weighted by population size to estimate the fraction of institutional traits among all communities. For simplicity, we refer to this measure as the “reach of rules” (i.e., the extent of their contact with the population).

We calculated the mean reach of administrative rules across all communities as follows: $m=\sum_{i}p_{i}z_{i}$, where $p_{i}=n_{i}/n$ is the relative membership size $n_{i}$ of community *i* over the total active population, n, among all communities, and $z_{i}$ is the relative frequency of administrative rules over all rules in community *i* (see Figure 1c).

The fitness of the reach of administrative rules (tracked with the letter m) is therefore dependent on the differential growth of certain kinds of rules (tracked with $z_{i}$) and the population size of the respective community that uses those rules (tracked with $p_{i}$). The Price equation decomposes the change in the reach of rules (change in m) (1) into how much the presence of such rules covaries with the growth of the population (a strong positive covariance would detect that more of this kind of rule goes together with an increase in the population size) and (2) into all remaining stochastic fluctuations observed according to the number of this kind of rule. This allows us to quantify how strongly the presence of a certain kind of rule covaries with changes in population size.

In this sense, we considered two factors regarding the frequency of change in administrative rules. The first is dependent on the differential membership population growth (fitness) associated with the different share of administrative rules within each community. This is based on the covariance term of the Price equation. In the context of Minecraft, we aimed to covary the differential membership growth rate in a community with a specific share of administrative rules. For example, if administrative rules are related to a higher population growth rate within a community, we may see that communities that implement exclusively administrative rules have a higher-than-average population reach growth. Several mechanisms may be behind this relationship. We cannot know the exact reasons of why this happens; perhaps communities with a large share of administrative rules are more likely to cultivate a good environment to attract more visitors, or perhaps an increased population requires a higher (or lower) proportion of administrative rules to manage public goods.

We now introduce a new variable, w, to track the rate of change of population shares, where $p_{i}^{\prime }$ is the proportion of administrative rules in the next time interval.
(1)$$p_{i}^{\prime }=p_{i}\frac{w_{i}}{w}$$

The mean rate of change will then be $w=\sum_{i=1}p_{i}w_{i}$. When the size of a population increases, the relative reach of the applied rules increases. In Minecraft communities, community goals are to survive and to recruit and retain more members. Accordingly, $\frac{w_{i}}{w}$ tracks the relative fitness change of communities. This source of change in the reach of the administrative rules can thus be seen as fitness-related change which may be tracked by expanding Equation (1) with the static number of rules: (2)$$p_{i}^{\prime }z_{i}=p_{i}\frac{w_{i}}{w}z_{i}$$

When the increase rate of the population is greater than the mean growth rate (i.e., $\frac{w_{i}}{w}$ > 1), the population reach of administrative rules in community *i m_i_* increases without any change in the frequency of administrative rule changes within community $z_{i}$. This fitness-correlated process is conceptually equivalent to selection acting at the scale of organizations.

It is worth noting that this is not a perfect replicator at the individual level in the Minecraft context because communities may have started (birth) or gone offline (die) during the time period in which we collected data. However, at the group level, when new communities come online and copy successful groups, the fraction of active populations that are constrained by the same rule strategy increases. When communities go offline, they lose their share of the population that is governed by the applied rule strategy. Additionally, they fail to provide a source for other communities to copy and thus, reduce the population share, which is subject to these types of rules. Thus, even without a perfect individual-level replicator, the cumulative institutional changes at the group level remain the same.

The second source of change in the weighted frequency of administrative rules is stochastic fluctuation, which may arise from drift [7] or transmission errors [59]. In this case, the following equation may be used:(3)$$p_{i}z_{i}^{\prime }=p_{i}\left(z_{i}+\delta_{i}\right)$$
where $\delta_{i}$ is some small random change in the frequency of administrative rule changes and $z_{i}^{\prime }$ indicates the same frequency within community *i* in the next time interval. 

Equations (2) and (3) can then be combined to simultaneously account for both selective and stochastic forces operating over time:(4)$$p_{i}^{\prime }z_{i}^{\prime }=p_{i}\frac{w_{i}}{w}(z_{i}+\delta_{i})$$

Change occurs both in the population size of each community and in the frequency of change of administrative rules. Using the above rules, it is straightforward to derive the Price equation (Price, 1970).
$$\begin{array}{c} \Delta m=m^{\prime }-m\\ =\sum_{i=1}p_{i}\frac{w_{i}}{w}\left(z_{i}+\delta_{i}\right)-\sum_{i=1}p_{i}z_{i}.\\ =\sum_{i}p_{i}\frac{w_{i}}{w}z_{i}-\sum_{i}p_{i}z_{i}+\sum_{i}p_{i}\frac{w_{i}}{w}\delta_{i}\\ =\sum_{i}p_{i}\frac{w_{i}}{w}z_{i}-\sum_{i}p_{i}\frac{w_{i}}{w}\sum_{i}p_{i}z_{i}+\sum_{i}p_{i}\frac{w_{i}}{w}\delta_{i} \end{array}$$

Price recognized that the first and second terms are equal to $E\left[\frac{w_{i}}{w},z_{i}]-E[\frac{w_{i}}{w}]E[z_{i}\right]$, which is the covariance between $\frac{w_{i}}{w}$ and $z_{i}$. The third term can be rewritten as $E\left[\frac{w_{i}}{w},\delta_{i}\right]$. However, since fluctuation $\delta_{i}$ has no correlation with $\frac{w_{i}}{w}$, the third term can also be written as $E\left[\delta_{i}\right]$. The equation can thus be simplified to
(5)$$\Delta m=COV\left[\frac{w_{i}}{w},z_{i}\right]+E\left[\delta_{i}\right]\;$$

The Price equation derived (i.e., did not postulate) a covariance between relative population growth $\frac{w_{i}}{w}$, and the relative frequency of a certain kind of rules, $z_{i}$. If this covariance is positive, it tells us that the presence of a certain kind of rule goes together with population growth. Since there is no intrinsic directionality in a covariance, this can be interpreted in two ways: the presence of a certain kind of rule helps the population to grow; or the growth of the population increases the reach of such rules. Both help to increase the reach of the rules within the total population among all communities (as measured by Δm).

The totality of the equation implies that selective forces (the first term) and stochastic forces (the second term) contribute to the frequency of change of institutional traits, Δz [13]. For empirical application, this can be reformulated as
(6)$$\Delta m=\beta VAR\left[z_{i}\right]+E\left[\delta_{i}\right]$$
where the coefficient *β* is the relative population growth $\frac{w_{i}}{w}$ based on the frequency of administrative rules $z_{i}$, and the var [$z_{i}$] is the total variance of administrative rules $z_{i}$ among all communities. In this equation, the slope reflects the strength of selective forces, and the intercept represents the strength of stochastic forces [58]. 

The logic behind the equation can be understood on two levels. At the individual community level, when a community first goes online, it needs to install rules from a pool of plugins. The community administrator can learn a governance style from other successful servers (success-biased learning) or apply a popular governance style (frequency-biased learning). The administrator can also try out new governance styles or new plugins developed in the Minecraft community (mutations) and can learn from other resources (individual learning). The result is that some of these implementations will be beneficial for the community to survive longer, for the governance style to be retained for longer, or for the community to be more successful so that other administrators will be more likely to learn from them. It could also be the case that the type of rules the community installs are detrimental to that community’s success, and thus, will lead to a shorter lifespan or a reduced likelihood of imitation by other communities. At the population level, the average share of rules that are beneficial for a community’s survival and success will increase due to the communities that are sustained in the population or social learning mechanisms. 

In an organizational context, the process described here may seem overly simple and abstract, especially considering other variables that may cause membership population increases or community death. Nonetheless, the Price equation describes the system of institutional evolution in a minimal manner and offers a way to identify and quantify selection- or fitness-correlated rule changes [60]. A fitness-correlated process is conceptually equivalent to selection acting at the scale of organizations. It is also worth noting that although we constructed those processes in our model, we did not establish causal links between the frequency of rule changes and active population share change. Instead, empirical estimations helped us to establish a connection between the two. 

As a result, the final linear equation demonstrates a relationship between rule frequency, variation, and selection: different types of rules have variations regarding their correlation with population growth. When the relationship of variation to total frequency changes (beta) is negative, it indicates that the administrative rule has a negative correlation with community membership growth. When communities have many variations between each other regarding the proportion of administrative rules, higher variation provides sources for selection and thus leads to higher frequencies of change, whereas when there is less variation, competition will be tight, and rates of change will be slower. A positive slope indicates a positive correlation between rule change frequency and population growth. This suggests that over time, communities with higher fractions of administrative rules will see an increased reach of administrative rules. Similarly, a negative slope indicates selection against administrative rules, suggesting that a higher proportion of administrative rules will decrease population reach over time.

### 2.3. Bet-Hedging and Information Theory

We pursued an inductive approach with the Price equation to estimate the strength of selection from frequency change. This enabled us to estimate the fitness-related frequency change of a particular rule through coefficient *β*. However, this approach also forced the growth rate to be fixed over time. Therefore, an average relative growth rate was assumed for each group of rules. This modeling strategy was found to be reasonable in a stable environment, but when the environment changed (see Figure 2), it did not always reveal the true dynamics of the frequency of change for two reasons: First, the selection forces on one type of rule depend not only on how well this rule performs on average (algorithmic mean of fitness-related growth) but also on whether implementing this rule instead of other rules led to community failure at a time of vulnerability (geometric mean of growth) [61]. The Price equation uses the arithmetic mean and thus cannot provide explanations for changing environments. Second, the selective forces on communities depend not only on the quantity of rules that covary positively with population growth but also on the rule combination strategy. It could be that some rules are beneficial in some periods, while others are useful in other periods. The average over all periods might suggest that it would be useful to favor one rule over another; however, eliminating the other rule entirely might crash the population in certain periods. In the Price equation, to evaluate the choice of one type of rule, we subsumed the influence of other rules within our stochastic forces residual, *E*[$\delta_{i}$]. This allowed us to focus on the dynamics of a single type of rule; however, it did not adequately explain how different types of rules can work together to assist communities in changing environments in a synergistic manner over time. 

One method that can incorporate the two factors not addressed in the Price equation is bet-hedging. Evolving biological and socio-economic populations can sometimes increase their growth rate by cooperatively redistributing resources among members. In unchanging environments, this simply comes down to reallocating resources to fitter types [62]. This would suggest that rules that are not useful for some time will get eliminated by natural selection. However, they might become useful again, and their premature elimination would then reduce fitness. Whenever there is a repeating cycle or seasonality in the fluctuations of an environment, it is useful to restrict the forces of blind natural selection during certain periods in order to be prepared for subsequent periods [59]. For example, it would not be useful to allow natural selection to eliminate all food storage during summer, even if ‘blind’ natural selection cannot see its utility during such periods. Neglecting predictable seasonality and merely working with cross-seasonal averages might suggest that the contribution of the “food storage rule” is, on average, negligible relative to the “go out and forage” rule and might suggest its elimination; however, this would cause the population to starve during winter. Any kind of anti-cyclical governmental policy or temporal economic subsidy exploits the same logic, i.e., of maximizing overall fitness in predictably fluctuating environments by combining different kinds of institutional mechanisms [59].

The utility of such portfolio theory-based bet-hedging over time depends on the predictability of the environmental patterns. If the future cannot be predicted, there is nothing to prepare for. However, if a predictable seasonality is known, one can adjust for it. The degree of predictability depends on the amount of information in the pattern, which leads us to information theory. 

In technical terms, if information about a future pattern is completely unrelated to the state of the environment, then the quantity of information common to the cue and the environment is zero. The cue does not help to increase the ‘fit’ between the evolving population and the environment. At best, a perfectly informative cue would precisely reveal the state of the environment: the remaining uncertainty about the environment would be zero and the population could be adjusted to grow optimally. This can be formalized using information that is common to the environment and the evolving population: the more you know (about the environment), the more you can grow (the population) [63].

This problem is formalized by portfolio theory and is the basic idea behind bet-hedging. It uses information about the environment to maximize long-term increase rates [64]. Built upon Kelly’s idea and subsequent expansions [65,66,67,68], Hilbert derived a measure to establish a cooperative resource redistribution strategy to maximize socioeconomic growth [63]. By establishing fitness matrices of rules for different environmental states, we can determine the most efficient rule distribution strategy for the sustainability and incrementation of this type of rule. Below, we apply the bet-hedging method to Minecraft.

First, we need to understand how the environment changes over time. The most efficient way to benefit from the changing dynamics in the environment is represented using information theory. Information, by definition, is related directly to the reduction of uncertainty. The ‘mutual information’ between a cue about an environmental pattern and a random environmental state measures how much the cue reduces uncertainty about the state and can be directly translated into growth potential [67]. It turns out that the pursuit of optimal growth consists of searching for mutual information (or unequivocal signals) between the environment and an evolving pattern [62].

In the case of Minecraft, if we already know the unfolding dynamics of environmental changes, we then have no uncertainty about the environment, i.e., we have perfect information with which to predict the state of the environment at any given time. In an ideal world, we can quantify how well one type of rule works in different environmental states and make confident decisions about implementing or removing this type of rule accordingly. However, in Minecraft, we have a high level of uncertainty about changes in the environment. What is the most efficient way to use information from previous events to assess the mechanisms of the current environment? Shannon answered this question by calculating the opposite side of information, i.e., uncertainty [69]. According to Shannon, the likelihood of encountering one specific environmental state from all possible states is equivalent to the reduced uncertainty caused by the knowledge of this specific state. We used a probability distribution describing whether the environment is friendly for one type of rule to represent the dynamic patterns of the environment. To minimize the number of assumptions required to measure the environment and establish one environmental measurement to consistently compare the changes of all four measures, we defined good and bad states based on whether the growth of the centralized rules (top-down administration, information broadcasting) outcompeted the growth of decentralized rules (communication, economics). This is aligned with hypotheses proposed by Perrow, i.e., that the paradox between centralization and decentralization grows with organizational complexity [70,71]. This cut-off allowed us to fit our data in a binary (computational) framework.

To optimize rule increments in a changing environment, we needed to redistribute the rule shares to match environmental-state probabilities. In extreme cases, rules that do not work well with the environment have a growth rate of 0. The optimal strategy is then to maintain the share of rule types in line with the corresponding environmental-state probabilities. Between extreme cases, we found that some rules in an unfavorable environment still yielded a positive growth rate. This required us to adjust the rule shares based on both the growth rate of different types of rules and the likelihood of different environmental states. Table 1 lists the growth rates of different types of rules in a good or bad state, where *W* refers to the growth rate in a good state and *w* refers to the growth rate in a bad state.

Solving for the optimal distribution of administrative rules sometimes results in undefined combinations of rule shares, including cases where the optimal share of rule *i* is negative (*d* < 0) or above 1 (*d* > 1). When *d* < 0, it suggests that the optimal strategy is to implement other rules only. Accordingly, for *d* > 1, the optimal strategy is to implement rule *i* only. These two extreme cases describe the so-called “pure strategy”. If the optimal strategy for rule *i* is a pure strategy of full investment in rule *i*, it indicates that (1) the frequency of change to rule *i* can be attributed to earlier investment in that rule, as opposed to other rules. In other words, the frequency of change of rule *i* is driven solely by selection; and (2) environmental changes do not alter the growth rate of rule *i* [63].

An optimal solution with a rule share between 0 and 1 is called the “region of bet-hedging” [67], which suggests a mixed proportion of different rules. The optimal timing to take advantage of cooperation among rule types to outperform blind competitive selection depends on the shape of the fitness landscape [59]; the more complementary the fitness of types in different environmental states, the proportionally larger the potential benefit of strategic cooperation over competitive selection. If the optimal strategy for rule *i* is in the region of bet-hedging, which implies a rationale for implementing *p* rule *i* and 1 − *p* other rules, it indicates that (1) the frequency change under rule *i* can be attributed to the earlier implementation of *i* as well as other rules; and (2) the environmental changes alter the growth rate of rule *I*, which is why we have to use a mixed rule combination (resources) to deal with the environmental changes (risk over time; [72]).

## 3. Results

### 3.1. The Price Equation Result 

To match the rule data with membership data, we marked 23 timestamps to estimate rule changes over time. At each time point for each community, we measured the fraction of each rule type (*z_i_*) and membership size (*p_i_*). We then calculated the average rule proportion weighted by membership size (*m*) and variations of rule proportions among communities. We partitioned the Price equation into slope-intercept forms (Equation (6). In this equation, the slope reflected the strength of selective forces, and the intercept represented the strength of stochastic forces [58]. As shown in Figure 3, each data point referred to a timestamp associated with variations in rule fractions *VAR* [*z_i_*] and the average population reach, *m*. We estimated that communities with administrative rules face positive selective forces (*β_admin_* = 0.117, *p* < 0.001) and negative stochastic forces (*E* [*δ*]*_admin_* = −3.602, *p* < 0.01). This indicated that, on the one hand, administrative rules have a strong positive correlation with community success in terms of recruiting and maintaining members, resulting in a higher probability that this type of rule structure will be learned by other communities. In other words, this direct, fitness-related benefit contributes to the growth of administrative rules. On the other hand, when driven by stochastic factors, including a lack of information, cultural preference/resistance, path dependency, or individual learning, administrators tend to reduce the proportion of administrative rules, regardless of their positive correlation with community fitness.

We also found positive selection of information rules (*β*_admin_ = 0.147, *p <* 0.05, see Figure 4a), indicating that information rules are beneficial for community survival.

We did not find statistically significant selection (slope) or stochasticity (intercept) divergence from 0 in communication (see Figure 4b) and economic rules (see Figure 4c), indicating that the frequency change of communication and economic rules were not significantly different from 0.

### 3.2. Bet-Hedging Result

We used bet-hedging to validate the Price equation result and see how the combination of different rules contributed to the frequency of rule changes.

The direct result of bet-hedging showed that the optimal situation for administrative rules to increase in number was to implement administrative rules only (*d_admin_ =* 1; See Figure 5). In other words, larger numbers of administrative rules can be attributed solely to the earlier implementation of administrative rules. The Price equation suggests that the theoretically optimal strategy is equivalent to the end result of pure natural selection. As such, it is consistent with the Price equation result, i.e., that a positive selective force is the only reason for the increase in the number of administrative rules.

At the same time, for information, communication, and economic rules, the optimal share is within the region of bet-hedging (0 < *d* < 1; see Figure 5), indicating that environmental changes alter the growth rates of the three types of rules, resulting in optimal strategies of mixed rule combinations (resources) in response to the environmental changes (risks over time). For these rules, it is useful to keep natural selection in check, as it would overexpose the community to rules which are less favorable in certain kinds of recurring environmental states. There is a “complementary variety” concerning the suitability of these rules in different environmental periods [62].

Combining the Price equation results, the optimal rule combination for informational rule growth (*d_information_ =* 0.703) showed that although informational rules generally have a positive correlation with community survival and success, this correlation varies over time. In a period when the growth rate of informational rules is low, the share of other rules helps the community in difficult times. As for communication and economic rules, although they do not provide individual selective advantages, they can be subsidized to help communities during environmental changes (*d_communication_* = 0.146; *d_economics_* = 0.420). 

Overall, we found that the environment for administrative rules is a winner-take-all, selection-driven-type situation. In contrast, for the other three types of rules, institutional diversity drives rule increment instead of competition and selection. In the long term, it is beneficial overall to maintain a certain mix of rules, even against the elimination pressure exerted by natural selection.

## 4. Discussion

In this study, we used the Price Equation and the bet-hedging method to quantify and isolate the drives of rule frequency changes in online communities. According to the relationships that the Price equation articulates, we found positive selection forces over administrative and informational rules. At the same time, stochastic forces, including random trials and cultural preferences, were found to lead to a decrease in the reach of administrative rules. We did not find significant rule reach changes in informational and economic rules. The bet-hedging result of optimal rule share supported this result and provided an additional explanation for the stochasticity quantified through the Price equation. We found that increases in the number of administrative rules were only driven by positive selection, whereas increases in information, communication, and economic rules were driven by institutional diversity as well. 

These results allowed us to consider the environmental states of rules. Administrative rules are in an environmental state when competition and selection dominate institutional evolution, whereas, for other rules, diversity and cooperation are the keys to success.

### 4.1. Contributions and Implications

This study used evolutionary frameworks and models to explain institutional development. By using an evolutionary framework, we did not disregard “agency” in institutional changes, but emphasized that, in the long run, the agency itself becomes endogenous through iterated learning and the selection and reproduction of practices and beliefs. On this basis, we integrated theories from organizational studies and formal models from evolutionary biology to explain the macro dynamics based on first principles under a given set of conditions. The empirical application of the Price equation in this paper helped us quantify selection and stochasticity and thus answers one of the fundamental questions faced in organizational studies: Are rules implemented for their direct benefit or for other reasons?

Our approach combined the advantages of comparative studies and mathematical models to show the dynamics and reveal collective patterns of institutional evolution [24]. Through a comparative analysis of thousands of communities in the same Minecraft environment, we could control for the spillover effects of other social processes and focus on the frequency changes of rules. Using non-linear mathematical models, we assessed institutional development not as a moment of equilibrium but as an evolving system where changes emerge based on some first principles and stochastic processes. The use of bet-hedging models complemented the Price equation results, demonstrating a practical application of information theory to answer evolutionary questions. 

Additionally, our bet-hedging results showed the influence of environmental fluctuations on evolutionary processes and identified a path by which to determine the current environmental states of particular institutional traits. Our estimations of environmental states and their influences could provide valuable information for risk-avoiding and decision-making, especially when other variables are fixed or controlled. This approach loosens the fixed environment assumptions in evolutionary models and helps make more accurate predictions in uncertain and risky environments.

The bet-hedging results also contribute to the literature on institutional diversity in three ways. First, our results empirically support the hypothesis that institutional diversity is beneficial to organizational development due to the application of certain rules. Second, we were able to calculate the boundary conditions of environmental states and specific rules in which institutional diversity bring about maximum benefits. Third, we extended the theory of diversity by demonstrating that diversity does not only benefit the overall collective fitness [25] but also contributes to the growth of a single rule (trait).

Although we focused on an online community, our results, to some extent, could be generalized to real-world communities and provide implications for policymakers and practitioners. The empirical evidence in this paper suggests that in a fast-changing environment, institutional diversity can be helpful for organizations to build resilience. 

Overall, in this research, we joined the conversation regarding population ecology research being carried out by online communities [2,4,49,73,74] to further understand organizational development. Ecological and evolutionary thinking provides two approaches to understanding the frequency of change in organizations. In recent years, researchers in different disciplines have tried to bridge the two grand theoretical frameworks and produce more integrated models [75,76]. Our work applies evolutionary thinking to recent empirical developments and advances the development of integrated models and model selection in organizational studies.

### 4.2. Limitations

The Price equation is a powerful tool for explaining the macro patterns of a system; however, it does not provide direct causal inferences. This is because the equation is ultimately a tautology [14] that describes frequency change. Thus, although we were able to estimate the strength of selection, we could not determine what drives the frequency of change beyond selection. Anything not directly related to community fitness is considered to be stochastic in nature, which we could not explain through the applied model. At the same time, the Price equation, when applied to cultural and organizational evolution, is poorly suited for organizational activities. In this research, we did not have a perfect replicator of rule change mechanisms. Replicators are not necessary for cumulative, adaptive cultural evolution [77] and provide less accurate estimations and interpretations of models compared to biological evolution estimations. Additionally, we used GLM to estimate selection and stochasticity in order to guarantee the robustness of the estimator. However, we could not be sure that the applied method was the most efficient. It is still debatable which estimation method is the most effective to estimate the slope of selection.

Our application of the bet-hedging method assumed a fixed fitness matrix due to the limitation of the technique [64]. Limited by computation power, we could only assume a two-state environment and calculate the shares for binary rule categories. This limitation simplified reality and forced an arbitrary choice concerning the environmental state. In this paper, we used the relative growth of centralized rules (administrative and informational) as an indicator with which to evaluate environmental states. While this allowed us to answer the research question, such a categorization is still relatively arbitrary and less theoretical. In future work, we may introduce more context-based measures of the environmental state based on organizational theories. 

Finally, we studied fitness in terms of the reach of rules. We tracked this among Minecraft communities in order to assess the influence of their reach, which was a justifiable definition of ‘rule fitness’ (i.e., selfish rules propagate). However, it did not tell us anything about other utilities of the rules (e.g., the level of satisfaction of users, economic or entertainment benefits of rules, etc.). This additional step could be achieved by relating the reach of rules to other performance measures of communities, as can be done with structural equation modeling [78].

### 4.3. Future Work 

Our methods represent a first attempt to apply evolutionary thinking to institutional development. They also point out where to look in datasets when analyzing institutional development. The general contribution is that we show that it is possible to apply long-standing formal theories of evolutionary change to determine concrete aspects of institutional evolution. The digital footprint produced by online communities and organizations will allow researchers to advance such an approach to a more formal stage of empirical testing and quantification. Existing evolutionary frameworks from evolutionary biology, such as the Price equation and bet-hedging, allow researchers to calculate long-standing measures and interpret them within solid conceptual frameworks. 

The Price equation indicated where to look when analyzing influences other than selection. For future research, we may want to use this information to look into the factors influencing stochastic forces. At the same time, the bet-hedging method points out where to look to identify the efficiency of institutional diversity in a changing environment. Future research may narrow the scale of institutional analyses to particular periods and rule shares in order to identify institutional effects. Future research can also look into the reasons for altering particular rules.

To summarize, this research describes the application of evolutionary thinking in institutional analysis and embraces the potential, provided by digital trace data, of macro-scale longitudinal analyses of online communities. By applying evolutionary models empirically, we are now able to quantitatively answer fundamental questions about institutional evolution and to open the door for future study of institutions from an evolutionary perspective.

## Figures and Tables

**Figure 1 entropy-24-01185-f001:**
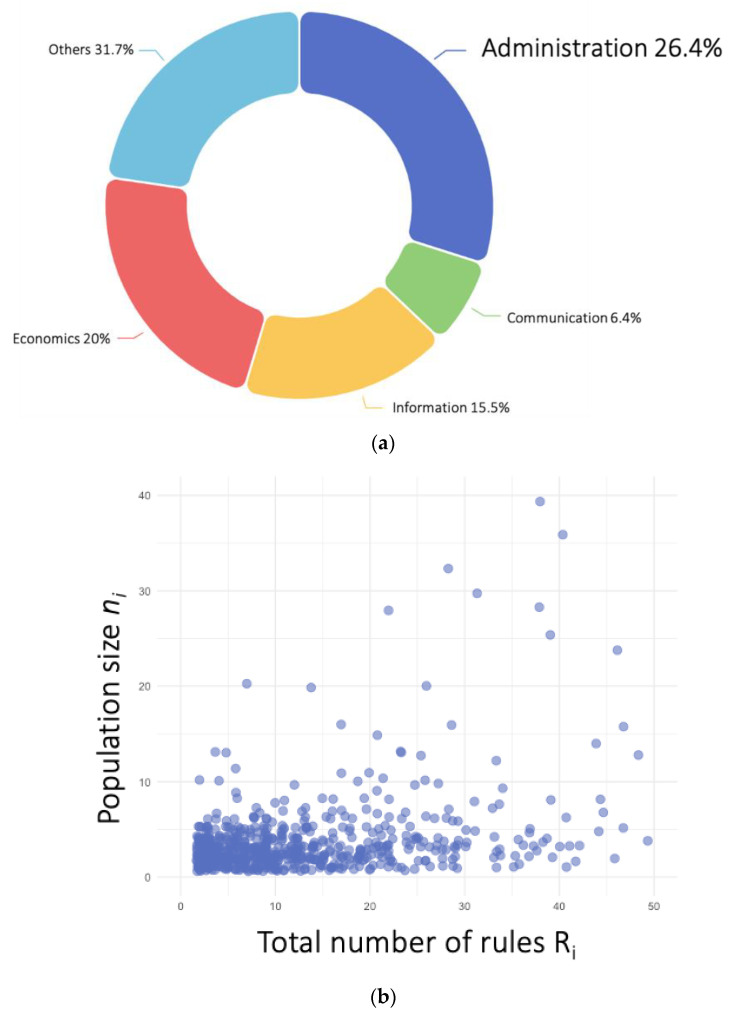
Model setup (**a**) Rule share pie chart of a community *i* at time *t:* each community has a fraction of administrative rules ($z_{i}$); (**b**) Membership size ($p_{i}$ ) by total rule number ($n_{i}$) scatter plot at time *t;* (**c**) the histogram of the administrative rule fraction ($z_{i}$) changes from time *t* to time *t* + 1, which also changes the average population reach *m*.

**Figure 2 entropy-24-01185-f002:**
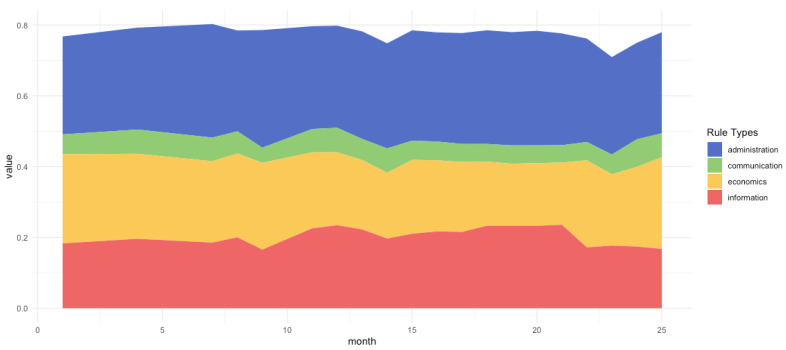
Environmental changes over time cause changes in the number of communities but do not seem to change the overall relative proportions of rule types in the population, except for administrative rules. The Price equation assumes a constant correlation between population growth and the implementation of one type of rule. However, the correlation may vary over time in a fast-changing environment. The changing bandwidth of administrative rules in this figure demonstrates that various rules are influenced differently by environmental changes.

**Figure 3 entropy-24-01185-f003:**
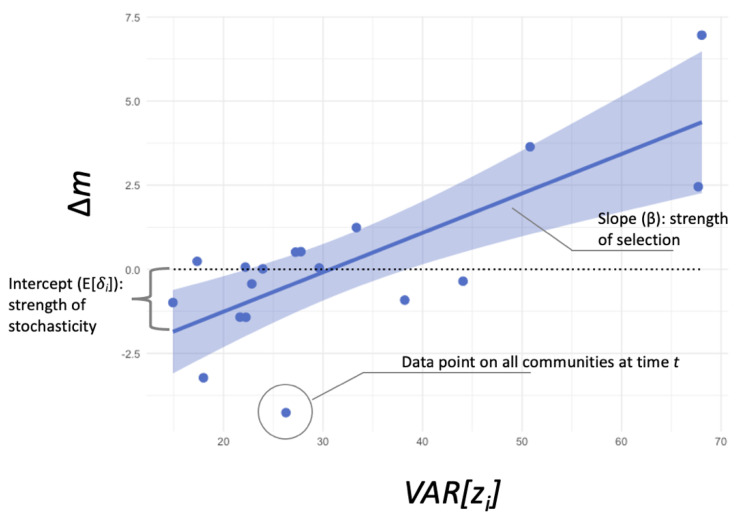
Communities subject to administrative rules face positive selective forces and negative stochastic forces. Administrative rules have a positive correlation with community fitness, which leads to a higher likelihood of this type of rule structure being learned by other communities. This direct fitness-related benefit is associated with the growth of administrative rules. On the other hand, other “stochastic” forces, including a lack of information, cultural preferences, cultural resistance, and random experiments, reduce the implementation of administrative rules.

**Figure 4 entropy-24-01185-f004:**
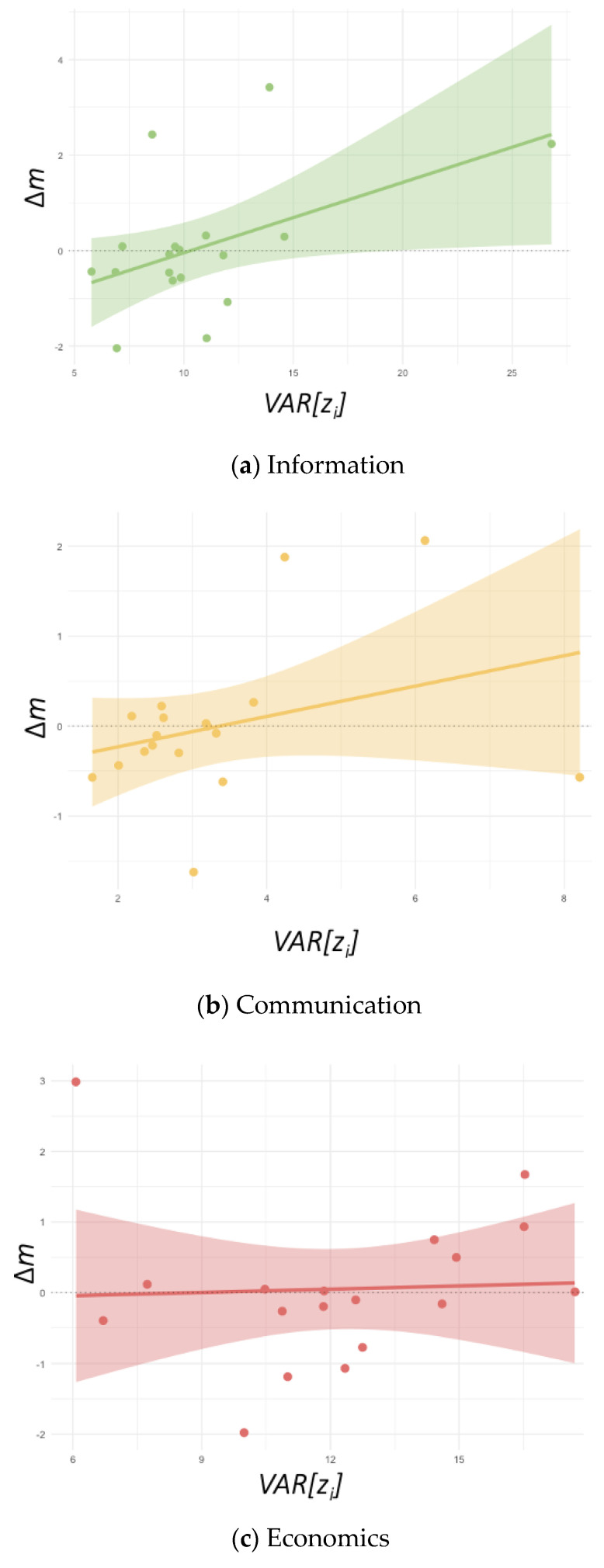
Communities with informational rules experience positive selective forces, while there are no effects of communication and economic rules on community prevalence. We found positive selection over informational rules but not negative stochastic forces (**a**). At the same time, neither selection (the slope) nor stochasticity (intercept) in communication (**b**) and economic rules (**c**) diverged significantly from 0 over time.

**Figure 5 entropy-24-01185-f005:**
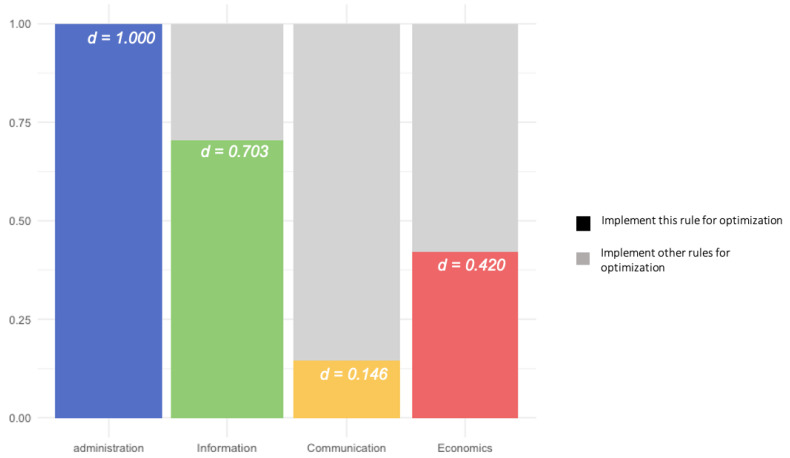
Most rules showed a maximum in their selective effect in combination with the other rule types. The bars in the figure illustrate the optimal distribution of rule implementation to maximize the growth rate of one type of rule, demonstrating the influence of implementing other types of rules on this type. For information rules to be expressed at a maximum rate in the population, the calculation suggests that they should be implemented with a 30% mix of other rule types. (This is distinct from the question of whether that maximum is positive, i.e., whether information rules are positively selected for, as shown in Figure 3). Implementing a mix of rules can help communities survive periods when the direct benefits of information rules are low. As a result, institutional diversity contributes to the long-term growth of communication, information, and economic rules. The optimal distribution of administrative rules, i.e., 100%, suggests an absolute strategy for the growth of this dominant rule type. This may be an artifact of the strong positive selection that communities with administrative rules face, particularly relative to other rule types. It is also consistent with the conclusion that the correlation between administrative rules and community fitness does not vary as much over time as it does with other rule types.

**Table 1 entropy-24-01185-t001:** Rule growth rate according to environmental state and rule categories.

State	Growth of Rule *i*	Growth of Other Rules
Good state for centralized rules	G_1_	g_1_
Bad state for centralized rules	g_2_	G_2_

## Data Availability

Data used in this study can be found in https://github.com/qkzhong/mc_Price_equation/blob/main/mc_dataset.csv (accessed on 28 April 2022).

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
