# Peer review of "Quantifying the Selective, Stochastic, and Complementary Drivers of Institutional Evolution in Online Communities"

_entropy, 2022, doi:10.3390/e24091185_

Round 1

Reviewer 1 Report

Thank you for the opportunity to review this manuscript. The authors provide a very innovative and impressive analysis of Minecraft data to demonstrate how the Price equation can shed light on the evolutionary processes responsible for organisational change. I found their methods to be robust, and they have a clear grasp of the limitations of their analysis. I believe that this manuscript represents a meaningful contribution to this outlet, and I do not hesitate to recommend it for publication.

My comments are purely editorial – I think that attending to these issues will only benefit their paper further:

On page 2, the authors appear to discuss their findings before they present them (e.g., line 88: “As a result, we found that administrative rules in Minecraft are beneficial for community success, and thus its frequency increases due to natural selection”). This is most likely a stylistic error, but if not, I would suggest reorganising the text to outline the theory and hypotheses in the introduction, and a summary of findings in the discussion.

The article would benefit from another edit. Some phrases are unclear (e.g., p. 3, line 130: “Second, group members can vote with feet and leave groups with poor institutions to join those with better institutions”), and there are consistent typographical / grammatical errors throughout.

Although the authors provide the Price equation later in the piece, for the benefit of the naïve reader, it might be helpful to introduce this when it is first discussed in Section 1.2.

Author Response

Thank you very much for the kind comments! We addressed the issues you pointed out as follows. For the first issue, we have shortened the specific findings and left two general sentences about our findings in the introduction to maintain the consistency of the structure.

For the second issue, we rephrase the sentence to “Group members have high leverage to migrate to communities with better institutions at a low cost” and fixed a few grammatical errors.

For the third issue, we added some introduction in section 1.2 to familiarize the audience.

Reviewer 2 Report

Review for "Is Institutional Evolution in Online Communities Driven by Selection or Stochasticity?"

In the manuscript, the authors seek to quantify the degree to which institutional change, here rules in Minecraft online communities, is controlled by an evolutionary process or bet-hedging.

I have a couple of conceptual problems with this approach.

The key data are about rules to be added or removed or changed in online communities, and the authors argue that their frequency change is intergenerational (also ref 1). And then an analogy to evolution is drawn. I have to point out that evolution is defined as a process requiring inheritance, variation, and selection. I fail to see how these abstract processes map to the data at hand. One needs an organism that copies, changes, and whose frequency changes.

But here: Is it the community that experiences inheritance? Is it the rule? What is the “organism”? This needs to be explained much better. Mainly because the authors state: “As a result, we found that administrative rules in Minecraft are beneficial for community success, and thus its frequency increases due to natural selection.” This suggests that it is the rule frequency that changes, making a rule analogous to “organism”. The results draw the same analogy and measure frequency changes of rules across online communities.

While I think it is implied, I now assume that each “rule” is an “organism”, and I will also ignore that evolution requires variation, which means a rule changes due to a mutation. Ironically, while a requirement for evolution, the authors ignore this, as, unfortunately, does the price equation … it deals with frequency changes, aka population dynamics, and not evolutionary dynamics, which includes the effect of mutations. However, it is a common simplification – a wrong one – but let's assume mutation rates are vanishing in this example.

What determines fitness is of course the fitness landscape. In the simplest form, it is a static match that assigns the potential mean number of offspring to each genotype/phenotype. This is really only observed in simple situations or genetic algorithms using a fitness function that only considers the performance of a single solution without considering other interactions. However, here it seems clear that the fitness of each rule of course depends on the other rules controlling each environment, and each environment (Minecraft community) experiences a different set of rules. Ergo, the fitness of each rule is dependent on the fitness (frequency) of other rules – aka frequency-dependent selection and inclusive fitness.

For a better analogy, imagine the Lotka-Volterra dynamics of foxes and bunnies – their frequency changes due to each other without extra evolutionary dynamics. Of course, applying the price equation to this system gives some results, but they would not be about evolution, but only about population dynamics. Here, the situation is more complicated, as the authors measure fox and bunny (well, really rule) frequencies across habitats. Which IMO makes it impossible to determine a fitness-based frequency change as this one depends on the frequencies of other rules, their interactions, and localized contexts.

The authors might argue that this complex interaction between rules doesn’t happen, or that the frequency of one rule does not affect the fitness of other rules … if that would be the case, we would look at a fitness landscape without epistasis which has no ruggedness and can be easily optimized by a hill climber. That would make the entire analysis trivial.

The authors discuss this limitation somewhat, but not conclusively.

Without evolution as a control, I am afraid, the results from the bet-hedging part are without context. The main idea of the experimental approach here is to say “is it A or B?”, but without A … statements about B (bet-hedging) again are without control.

The authors also discuss that the bet-hedging approach I limited to paired interactions, and as I argued before, here we are facing a more complex situation.

Unfortunately, I don’t see a good way to improve the manuscript. Measuring fitness, even with great data, particularly in a dynamically changing environment where interactions matter and entities experience mutations, is pretty much impossible to do.

Author Response

We thank the reviewers for the questions on the conceptual framework, and we acknowledge that we use Darwin’s approach differently from population biology. We explain here how inheritance and evolutionary forces work in this case.

The approach to understanding cultural and institutional organization is analogous to Darwin’s approach to analyzing organic evolution. The fundamental element was the assumption that organisms inherit information that determines their phenotype, in combination with the local environment. Assuming that culture is a system of inheritance, Darwin’s approach is useful in that it helps us focus on the processes that affect the variation carried through time by a succession of individuals.

We take a community as an organism and the share of rules (plugins) as the institutional trait or cultural variant it displays. In this large group of communities in Minecraft, we see the different communities exhibit different institutional traits (cultural variants), which constitute the overall distribution of institutional traits in Minecraft. To understand why a particular distribution of cultural variants characterizes the Minecraft environment, we must understand the forces of cultural evolution that act on each community.

Indeed, we are not sure about the actual inheritance process in Minecraft, and we do not know if after a community dies, the governing knowledge is inherited by its members to pass on to the next generation. So in this sense, we do not strictly follow a genetic inheritance model. However, when a new community starts, to maintain community survival and deal with collective action problems, the community administrators need to learn socially from other communities or learn independently from the environment to establish their institutional traits (governing style). The process of learning from other communities, either previously established communities or the communities at the same age, can be seen as the cultural transmission that increases or decreases the frequency of some kinds of institutional traits (governing style), and thus changes the overall distribution of rule shares. 

Different forces contribute to the changes in overall distributions. Selective forces act on Minecraft communities in two ways. First, communities that employ the governance style that contribute positively to community survival will last longer. For example, if administrative rules are the most beneficial for community survival, communities that employ a governance style with a large share of administrative rules will last longer. In contrast, communities that employ a governance style with a small share of administrative rules will die out faster. In our dataset, community death refers to the stage where servers are taken off the Internet. This differential survival rate of different governing styles will lead to a shift in the overall distribution of administrative rules. Second, communities that employ the governance style beneficial for community success are more likely to be copied by other communities. The spread of successful governance styles can also change the overall distribution of rule shares.

Stochastic forces include drift and other not payoff-related causes in Minecraft communities. Stochastic forces also act on Minecraft communities mostly in three ways. First, communities learn from other communities blindly. When the learning is not led by success bias but rather by proximity or uncertainty, this type of copying will lead to drift in the overall distribution. Second, when players cultivate cultural preferences of specific governance styles, they are likely to spread these specific rule shares to other communities they migrate to (Zhong & Frey, 2020).

Additionally, mutation provides additional variation for selection, and in Minecraft, the introduction of new plugins or individual learning to establish new governance styles can be seen as mutation.

Administrators do not know whether their decision-making in implementing one type of rule instead of other types of rules is due to selective or stochastic forces. At the same time, we cannot accurately identify from the interactions between communities the selective forces or stochastic forces, as well as most historical data on cultural evolution. What we can access from the data is the collective pattern of rule distribution. The advantage of the Price equation is that, in this case, it isolates the selective forces from the stochastic forces statistically without specifying a large number of possible mechanisms of each basic process (e.g., selection, mutation, drift).

We added some explanations to the Data section for a better understanding.

As for the binary cut-off in bet-hedging models, we admit that it is a somewhat arbitrary assumption to make in a complex environment but it is the least assumption we can make to model the environmental state we know very little about.

Author Response

We thank the reviewer for the thoughtful comments and suggestions. The detailed and constructive questions also help us to respond to other reviewers and improve the manuscript. We write in blue text our response and solution.

Reviewer 4 Report

The manuscript raises a highly challenging research question: Is the institutional evolution in online communities driven by selection or stochastic pathways?

Although the research question is interesting, it requires several improvements in line with what was clearly stated in my previous lines. Moreover, the literature review needs to be reinforced, especially the one concerning the Institutional Approach. The paper could provide implications both for policy-makers and practitioners concerning online communities.

Albeit it is recognised merit to the research developed so far, in order to improve the global quality of the manuscript it is recommended to improve the following aspects:

  1. Improve the motivation of the guiding research question, in the introductory section.
  2. Identify the contribution to the literature devoted to Entropy and Markets Evolution.
  3. Reinforce the literature review concerning the Institutional Theory recovering the classic works of Ronald Coase, Douglas North and Oliver Williamson, on institutions and transaction costs, and more recently, of Paul Belleflamme and Martin Peitz, on the Economics of Platforms.
  4. There is a need for justifying the use of a generalized linear model (GLM) specification.
  5. In the Conclusions section, there is room for providing implications both for policymakers and practitioners, bearing in mind the empirical evidence now obtained.

Author Response

Thank you for the suggestions. We tried to address most of them in the manuscript. Although we recognize the value of the literature and further discussion you suggested, given the time limit, we were not able to do an extensive reading for more meaningful incorporation of the rich literature you brought up.  

  1. Improve the motivation of the guiding research question, in the introductory section.

Response: We motivate the research question by pointing out a main question in organizational studies: what drives institutional changes in organizations? We now stressed this in the introduction to better motivate our research question.

2. Identify the contribution to the literature devoted to Entropy and Markets Evolution.

Response: Good point. We now mentioned in discussion that our method contribute to the application of information theory in evolutionary question, as the contribution to Entropy. However, our contribution to market evolution is limited.

3. Reinforce the literature review concerning the Institutional Theory recovering the classic works of Ronald Coase, Douglas North and Oliver Williamson, on institutions and transaction costs, and more recently, of Paul Belleflamme and Martin Peitz, on the Economics of Platforms.

Response: Thank you for the suggestions, we agree that the works of  Ronald Coase, Douglas North and Oliver Williamson are valuable to the framework and we now added these work in the introduction. However, the paper is not directly relevant to the economics of platforms. We use the platform of Minecraft not for the unique platform values but for the large-scale comparison it can provide for institutional analysis. So we decided not to incorporate it in the overall framework.

4. There is a need for justifying the use of a generalized linear model (GLM) specification.

Response: We partition the Price equation into slope-intercept forms. Provided that we can measure the change in mean frequency of rule types among different rule types, the strength of selection and stochastic forces thus can be estimated directly from rule data using linear models. Generalized linear models, given its loosened assumption on error distributions, can be used as an appropriate estimator for the selective and stochastic forces in this case.  It is worthnoting that Certain assumptions need to be met while using GLM in this case. First, it is important that the frequencies of rules are not be too close to the extremes. When the share of rules is too close to 0 or 1, there is no room for change. If the share of one type of rules is too close to 1, it is not free to change under selective forces. If the share of one type of rules is close to 0, it is not free to change under stochastic forces. Second, selective or stochastic forces should not be too extreme to overwhelm the capacity of GLM. Our data pattern meets these assumptions.

We addressed this in method section.

5. In the Conclusions section, there is room for providing implications both for policymakers and practitioners, bearing in mind the empirical evidence now obtained.

Response: Thank you. We now added implications for practitioners. We were a bit conservative about applying the online community empirical result directly to the real-world settings but we added that institutional diversity could be useful for real-world practitioners to consider

Round 2

Reviewer 2 Report

I raised substantial concerns about the applicability of the price equation, mostly because it is not clear how the rules controlling Minecraft communities actually map to evolving entities. The three fundamental principles inheritance, variation, and selection must be found in the system that should be investigated to prove or measure its evolution.

On top of that, does the Price equation deal primarily with frequency changes without mutations, and is thus the least relevant measure to show evolution.

The authors in their response - I paraphrase - redefine what evolution is to fit their data and do not properly map/align their system to evolution. The changes I would need to see to recommend publication are much more extensive than the authors provided. 

Author Response

We thank the reviewer for stressing the principle of evolution and encouraging us to clarify the mapping between the evolutionary process and institutional changes in online communities. Our responses are in the attached file in blue text.

Author Response

We have responded to your questions in blue text in the attached letter. Your questions have been helpful for us to specify the evolutionary process, improve the manuscript and answer other reviewers’ questions. We appreciate all of your detailed comments, questions and suggestions.

Reviewer 4 Report

After the changes, the global quality of the manuscript has increased.

Author Response

Thank you for the suggestions and help in your review!